# DEEP AUTOMODULATORS

## ABSTRACT

We introduce a novel autoencoder model that deviates from traditional autoencoders by using the full latent vector to independently modulate each layer in the decoder. We demonstrate how such an 'automodulator' allows for a principled approach to enforce latent space disentanglement, mixing of latent codes, and a straightforward way to utilize prior information that can be construed as a scale-specific invariance. Unlike the GAN models without encoders, autoencoder models can directly operate on new real input samples. This makes our model directly suitable for applications involving real-world inputs. As the architectural backbone, we extend recent generative autoencoder models that retain input identity and image sharpness at high resolutions better than VAEs. We show that our model achieves state-of-the-art latent space disentanglement and achieves high quality and diversity of output samples, as well as faithfulness of reconstructions.

## 1 INTRODUCTION

This paper introduces a new generative autoencoder for learning representations of image data sets, in a way that allows arbitrary combinations of latent codes to generate images (see Fig. 1). We achieve this with an architecture that uses adaptive instance normalization (AdaIn, Dumoulin et al., 2017b; Huang & Belongie, 2017), and training methods that let the model learn a highly disentangled latent space by utilizing progressively growing autoencoders (Heljakka et al., 2019). In a typical autoencoder, input images are encoded into latent space, and the information of the latent variables is then passed through successive layers of decoding until a reconstruction of the input image has been formed. In our model, the latent vector independently *modulates* the statistics of each layer of the decoder—the output of layer $n$ is no longer solely determined by the input from layer $n-1$.

In image generation, the probability mass representing sensible images (such as human faces) lies concentrated on a low-dimensional manifold. Even if impressive results have been shown for image generation (*e.g.*, by GANs, Goodfellow et al., 2014; Karras et al., 2019), efficient reconstruction and manipulation remain open problems. Deep generative autoencoders provide a principled approach for feature extraction, editing, and image synthesis. We show that modulation of decoder layers with AdaIn further improves these capabilities. It also yields representations that are more disentangled, a property here broadly defined as something that allows for fine control of one semantic (image) sample attribute independently of others. Previous works on AdaIn are mostly based on GAN models (Karras et al., 2019; Chen et al., 2019) with no encoder for new input images.

Autoencoders are typically single-pass encoder–decoder structures, where a sample enters from one end and is reconstructed at the other. For robustness, the reconstructed samples could be re-introduced to the encoder, repeating the process, and requiring consistency between the passes. In comparison to stochastic models (like VAEs, Kingma & Welling, 2014; Rezende et al., 2014), our deterministic model is better suited to take advantage of this. We can mix the latent codes of separate samples and measure the conservation of layer-specific information for each. This enforces disentanglement of layer-specific properties, because we can ensure that the latent code introduced from a certain layer onwards on the 1st pass should not affect the previous layers on the 2nd pass. Image autoencoders with convolutional architectures are often used for finding attributes that are disentangled from each other in the latent space. Often, they have relatively poor output image quality, and prior information must be fed in either via class-conditioning or by more complex loss functions for the latent variables. Fully implicit methods avoid this, but without a built-in encoding mechanism often cover only a relatively small fraction of the variation in the training data. For the first time, we show a full autoencoder architecture that has a state-of-the-art disentanglement

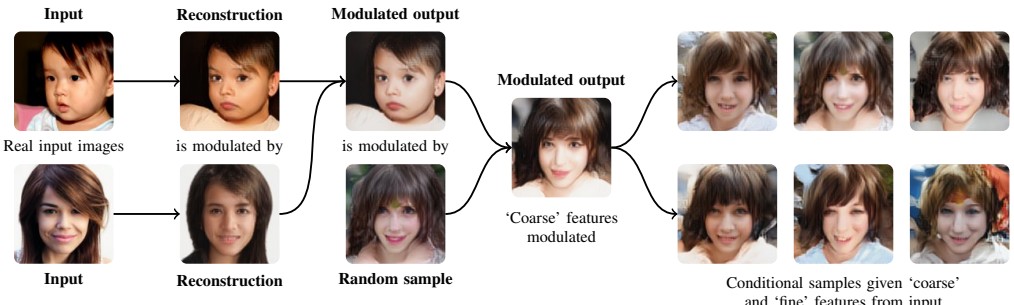

Figure 1: Illustration of some automodulator capabilities. The model can directly encode real (unseen) input images (left). Inputs can be mixed by modulating one with another or with a randomly drawn sample, at desired scales (center). Finally, taking random modulations for certain scales produces novel samples conditioned on the input image (right).

performance even for unsupervised training setup in several data sets, sharp image output quality, good coverage of the variation of the training data, arbitrary mixing of latent representations and a principled approach for incorporating scale-specific prior information.

Our contributions are as follows. *(i)* We introduce the *automodulator*, a new autoencoder-like model with powerful properties not found in regular autoencoders, including scale-specific style transfer (Gatys et al., 2016). In contrast to architecturally similar 'style'-based GANs, we can now directly encode and manipulate new inputs. *(ii)* We present methodology that allows *fully unsupervised* training of such a model to learn rich latent representation in high resolutions, and specifically explain a novel *multi-pass* training principle that enables the model to learn a more disentangled representation of training data than regular autoencoders. *(iii)* We demonstrate qualitative and quantitative performance and applications on CELEBA-HQ and LSUN Bedrooms and Cars data sets, taking advantage of the highly disentangled model structure.

## 2 RELATED WORK

Our work builds upon several lines of previous work in unsupervised representation learning. The most relevant concepts are variational autoencoders (VAEs, Kingma & Welling, 2014) and generative adversarial networks (GANs, Goodfellow et al., 2014). In VAEs, an encoder maps data points to a lower dimensional latent space and a decoder maps the latent representations back to the data space. The model is learnt by minimizing the reconstruction error together with a regularization term that encourages the distribution of latents to match a predefined prior. Latent representations often provide useful features for applications (*e.g.*, image analysis and manipulation) and decoding random samples from the prior allows novel data generation. However, in case of images, the generated samples are often blurry and not fully photorealistic, with imperfect reconstructions.

Current state-of-the-art in generative image modeling is represented by GAN models (Brock et al., 2019; Karras et al., 2019) which achieve higher image quality than VAE based models. Nevertheless, these GANs lack an encoder for obtaining the latent representation for a given image, limiting their usefulness. In some cases, a given image can be semantically mapped to the latent space via generator inversion but this iterative process is prohibitively slow for many applications, and the result may depend on initialization (Abdal et al., 2019; Creswell & Bharath, 2019).

Bidirectional mapping has been targeted by VAE-GAN hybrids such as Makhzani et al. (2016); Mescheder et al. (2017); Makhzani (2018), and adversarial models such as Donahue et al. (2017); Dumoulin et al. (2017a); Donahue & Simonyan (2019). These models learn mappings between the data space and latent space using combinations of encoders, generators and discriminators. In-troVAE (Huang et al., 2018) and AGE (Ulyanov et al., 2018) are more compact autoencoders, which both contain only two deep networks, encoder and decoder, learnt using a combination of reconstruction loss and adversarial loss, which encourages the distributions of real and generated images to match with the prior in latent space. A progressively grown version of AGE, PIONEER, was proposed by Heljakka et al. (2018; 2019). There is also recent work on vector quantized variational

autoencoders (VQ-VAE, Razavi et al., 2019) with a discrete latent space. However, while suitable for image compression, the discrete latent representation cannot *e.g.* be interpolated and hence may not be optimal for semantic image manipulation and learning a disentangled latent space.

The architecture of our generator and AdaIn utilization was inspired by the recent StyleGAN (Karras et al., 2019), but has no discriminator (nor the 'mapping network' $f$). Instead, our model simply contains the generator (*i.e.*, decoder) and an encoder, jointly trained applying a similar progressive growing strategy as Heljakka et al. (2018), but with a modified loss formulation. Besides the reconstruction loss and adversarial loss from Ulyanov et al. (2018) and Heljakka et al. (2018), we can also recirculate style-mixed reconstructions as 'second-pass' training samples in order to better encourage the independence and disentanglement of emerging styles and conservation of layer-specific information. The recirculation idea is biologically motivated and conceptually related to many works, such as that by Hinton & McClelland (1988). Utilizing the outputs of the model as inputs for the next iteration is related to, *e.g.*, Zamir et al. (2017), where feedback is shown to benefit image classification, and in RNN-based methods (Rezende et al., 2016; Gregor et al., 2015; 2016).

## 3 METHODS

We now describe the model architecture, starting from regular PIONEER and building up to the automodulator (Sec. 3.1). We then describe the unsupervised training via our extensions to the regular PIONEER loss function, to support flexible mixing of latents (Sec. 3.2). Finally, we introduce an additional weakly supervised training method to utilize known invariances (Sec. 3.3).

### 3.1 AUTOMODULATOR ARCHITECTURE

Our interest is in unsupervised training of an autoencoder wherein the inputs $\boldsymbol{x}$ are images fed through an encoder $\boldsymbol{\phi}$ to form a low-dimensional latent space representation $\boldsymbol{z}$ (we use $\boldsymbol{z} \in \mathbb{R}^{512}$, normalized to unity). This representation can then be decoded back into an image $\hat{\boldsymbol{x}}$ through a decoder $\boldsymbol{\theta}$. For stable training, we adopt the progressively growing architecture of Balanced PIONEER (Heljakka et al., 2018; 2019). The convolutional layers of the symmetric encoder and decoder are faded in gradually during the training, in tandem with the resolutions of training images and generated images (Fig. 2).

**Adaptive Instance Normalization (AdaIn)** A traditional decoder architecture would start from a small-resolution image and expand it layer by layer until the full image is formed, feeding the full information of the latent code via the decoder layers. In contrast, our decoder is composed of layer-wise functions $\boldsymbol{\theta}_i(\boldsymbol{\xi}_{i-1}, \boldsymbol{z})$ that separately take a 'canvas' variable $\boldsymbol{\xi}_{i-1}$ denoting the content input from the preceding decoder layer (see Figs. 2 and 3a), and the actual (shared) latent code $\boldsymbol{z}$. First, each deconvolutional layer #i computes its feature map output $\boldsymbol{\chi}_i$ from $\boldsymbol{\xi}_{i-1}$ as in traditional decoders. Second, we take the AdaIn normalization step that uses $\boldsymbol{z}$ to modulate the channel-wise mean $\mu$ and variance $\sigma^2$ of the output. To do this, we need a map $g_i : \boldsymbol{z} \mapsto (\boldsymbol{\mu}_i, \boldsymbol{\sigma}_i^2)$, arriving at:

$$\boldsymbol{\xi}_i = \text{AdaIn}(\boldsymbol{\chi}_i, g_i(\boldsymbol{z})) = \boldsymbol{\sigma}_i \left( \frac{\boldsymbol{\chi}_i - \mu(\boldsymbol{\chi}_i)}{\sigma(\boldsymbol{\chi}_i)} \right) + \boldsymbol{\mu}_i. \tag{1}$$

We implement $g_i$ for layer #i as a fully connected linear layer, with output size $2\,C_i$ for $C_i$ channels. Layer #1 starts from a constant input $\boldsymbol{\xi}^{(0)} \in \mathbb{R}^{4 \times 4}$. Here, we focus on pyramidal decoders with monotonically increasing resolution and decreasing number of channels, but any deep decoder would be applicable. The AdaIn-based architecture allows the output of layer #i to be not solely determined by the input from layer #$i-1$, enabling finer control over the image information, image mixing schemes (Sec. 3.2), and utilization of scale-specific invariances in training data (Sec. 3.3).

### 3.2 STRONG CONSERVATION OF CYCLIC INFORMATION: LAYER-SPECIFIC LOSS METRICS

In training of regular AGE and PIONEER models, the encoder $\phi$ and the decoder $\theta$ are trained at separate steps, optimizing either the loss $\mathcal{L}_\phi$ or $\mathcal{L}_\theta$, correspondingly:

$$\mathcal{L}_\phi = \text{D}_{\text{KL}}[q_\phi(\boldsymbol{z} \,|\, \boldsymbol{x}) \,\|\, \text{N}(\boldsymbol{0}, \mathbf{I})] - \text{D}_{\text{KL}}[q_\phi(\boldsymbol{z} \,|\, \hat{\boldsymbol{x}}) \,\|\, \text{N}(\boldsymbol{0}, \mathbf{I})] + \lambda_{\mathcal{X}} \, d_{\mathcal{X}}(\boldsymbol{x}, \boldsymbol{\theta}(\boldsymbol{\phi}(\boldsymbol{x}))),$$

$$\mathcal{L}_\theta = \text{D}_{\text{KL}}[q_\phi(\boldsymbol{z} \,|\, \hat{\boldsymbol{x}}) \,\|\, \text{N}(\boldsymbol{0}, \mathbf{I})] + \lambda_{\mathcal{Z}} \, d_{\cos}(\boldsymbol{z}, \boldsymbol{\phi}(\boldsymbol{\theta}(\mathbf{z}))), \tag{2}$$

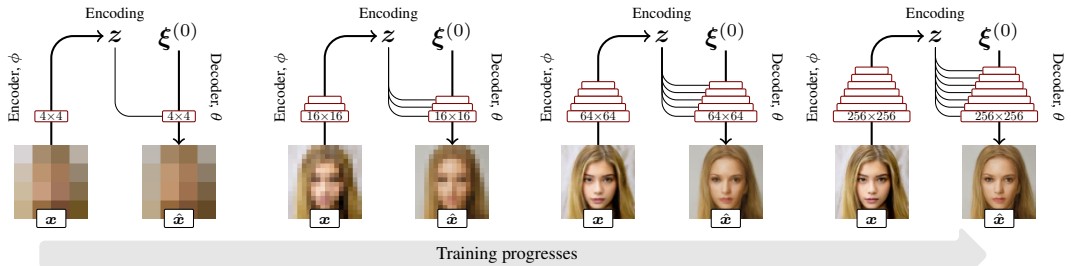

Figure 2: The model grows step-wise during training; the resolution doubles on every step. Input $\boldsymbol{x}$ is encoded into a latent enocoding $\boldsymbol{z}$ (a dimensionality of 512 used throughout this paper). The decoder acts by modulating an empty canvas $\boldsymbol{\xi}^{(0)}$ by the latent encoding and produces the output $\hat{\boldsymbol{x}}$.

where $\boldsymbol{x}$ is sampled from the training set, $\hat{\boldsymbol{x}} \sim q_\theta(\boldsymbol{x} \,|\, \mathbf{z})$, $\boldsymbol{z} \sim \mathrm{N}(\mathbf{0}, \mathbf{I})$, $d_\mathcal{X}$ is L1 or L2 distance, and $d_{\cos}$ is cosine distance. Since the model allows sampling from latent space (similarly to VAEs and GANs), the KL divergence terms during training can be calculated from empirical distributions, despite the model itself being fully deterministic. Yet we retain, in principle, the full information about the image, at every stage of the processing. For an image encoded into a latent vector $\boldsymbol{z}$, decoded back to image space as $\hat{\boldsymbol{x}}$, and re-encoded as latent vector $\boldsymbol{z}'$, it is possible and desirable to require that $\boldsymbol{z}$ is as close to $\boldsymbol{z}'$ as possible, yielding the latent reconstruction error $d_{\cos}(\boldsymbol{z}, \boldsymbol{\phi}(\boldsymbol{\theta}(\mathbf{z})))$.

**Layer-specific Loss** We now show how to encourage the latent space to become hierarchically disentangled with respect to the levels of image detail, allowing one to separately retrieve 'coarse' and 'fine' aspects of a latent code. This allows for *e.g.* conditional sampling by fixing the latent code for specific layers of the decoder, or mixing the scale-specific features of two or more input images—impossible feats for a traditional autoencoder with mutually entangled decoder layers.

The latent reconstruction error as such can be trivially generalized to any layer of $\boldsymbol{\theta}$. We simply pick the layer of measurement, and from there, pass the sample once through a full encoder-decoder cycle. But now, in the automodulator, latent codes can be introduced on a per-layer basis, enabling more powerful reconstruction measurements. (Without loss of generality, here we only consider mixtures of 2 codes.) We can present a decoder (Fig. 3b) with $N$ layers, split after the $j^{\text{th}}$ one, as a composition of $\boldsymbol{\theta}_{j+1:N}(\boldsymbol{\theta}_{1:j}(\boldsymbol{\xi}^{(0)}, \boldsymbol{z}_A), \boldsymbol{z}_B)$. Crucially, we can choose $\boldsymbol{z}_A \neq \boldsymbol{z}_B$ (extending the method of Karras et al., 2019), such as $\boldsymbol{z}_A = \boldsymbol{\phi}(\boldsymbol{x}_A)$ and $\boldsymbol{z}_B = \boldsymbol{\phi}(\boldsymbol{x}_B)$ for (image) inputs $\boldsymbol{x}_A \neq \boldsymbol{x}_B$. Because the earlier layers #1:$j$ operate on image content at lower resolutions, the output fusion image mixes the 'coarse' features of $\boldsymbol{z}_A$ with 'fine' features of $\boldsymbol{z}_B$. Now, $\boldsymbol{z}$ holds feature information at different levels of detail, some of which are mutually independent. Hence, when re-encoding an image, we should keep the representation of those levels disentangled in $\boldsymbol{z}$, even if they originate from separate source images. Hence, when we re-encode the fusion image into $\hat{\boldsymbol{z}}_{AB}$, and decode once more, the output of $\boldsymbol{\theta}_{1:j}(\boldsymbol{\xi}^{(0)}, \hat{\boldsymbol{z}}_{AB})$ should be unaffected by $\boldsymbol{z}_B$.

This leads to the following conjecture. Assume that the described network reconstructs input samples perfectly, *i.e.* $\boldsymbol{x} = \boldsymbol{\theta}(\boldsymbol{\phi}(\boldsymbol{x}))$. Any $\boldsymbol{z}_A$ and $\boldsymbol{z}_B$ can be mixed, decoded and re-encoded as

$$\hat{\boldsymbol{z}}_{AB} \sim q_\phi(\boldsymbol{z} \,|\, \boldsymbol{x}) \, q_{\theta_{j+1:N}}(\boldsymbol{x} \,|\, \boldsymbol{\xi}^{(j)}, \boldsymbol{z}_B) \, q_{\theta_{1:j}}(\boldsymbol{\xi}^{(j)} \,|\, \boldsymbol{\xi}^{(0)}, \boldsymbol{z}_A). \tag{3}$$

Now, between $\boldsymbol{\xi}^{(j)}$ of the first and $\boldsymbol{\xi}^{(j)}$ of the second pass (see Fig. 3c), the mutual information is

$$I[q_{\theta_{1:j}}(\boldsymbol{\xi}^{(j)} \,|\, \boldsymbol{\xi}^{(0)}, \hat{\boldsymbol{z}}_{AB}); q_{\theta_{1:j}}(\boldsymbol{\xi}^{(j)} \,|\, \boldsymbol{\xi}^{(0)}, \boldsymbol{z}_A)]. \tag{4}$$

With $\boldsymbol{z}_A, \boldsymbol{z}_B \sim \mathrm{N}(\mathbf{0}, \mathbf{I})$, for each $j$, we can now maximize (4) by minimizing

$$\mathcal{L}_j = d(\boldsymbol{\theta}_{1:j}(\boldsymbol{\xi}^{(0)}, \hat{\boldsymbol{z}}_{AB}), \boldsymbol{\theta}_{1:j}(\boldsymbol{\xi}^{(0)}, \boldsymbol{z}_A)) \tag{5}$$

for some distance function $d$ (here, L2 norm). In other words, the fusion image can be encoded into a new latent vector $\hat{\boldsymbol{z}}_{AB}$ in such a way that, at each layer, the decoder will treat the new code similarly to whichever of the original two separate latent codes was originally used there (see Fig. 3b). For a perfect network, $\mathcal{L}_j$ can be viewed as a layer entanglement error. Randomizing $j$ during the training, we can measure $\mathcal{L}_j$ for any layers of the decoder.

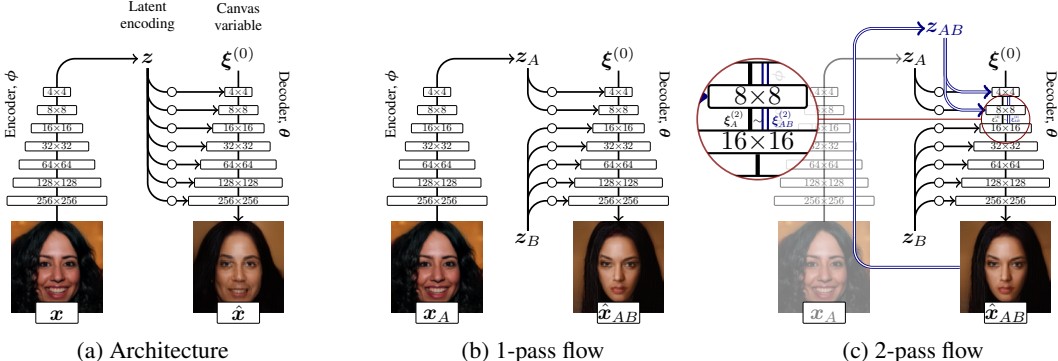

Figure 3: (a) The autoencoder-like usage of the model. (b) Modulations in the decoder can come from different latent vectors. This can be leveraged in feature/style mixing, conditional sampling, and during the model training (first pass). (c) The second pass during training.

**Final Unsupervised Automodulator Training Loss**   Our results improved by using a robust image reconstruction loss $d_\rho$ of Barron (2019) instead of L1/L2. $d_\rho$ generalizes various norms and exposes robustness as an explicit continuous parameter vector $\boldsymbol{\alpha}$. The complete loss functions are

$$\mathcal{L}_\phi = \max(-M_{\text{gap}}, \text{D}_{\text{KL}}[q_\phi(\boldsymbol{z} \,|\, \boldsymbol{x}) \,\|\, \text{N}(\mathbf{0}, \mathbf{I})] - \text{D}_{\text{KL}}[q_\phi(\boldsymbol{z} \,|\, \hat{\boldsymbol{x}}) \,\|\, \text{N}(\mathbf{0}, \mathbf{I})]) + \lambda_\mathcal{X} \, d_\rho(\boldsymbol{x}, \boldsymbol{\theta}(\phi(\boldsymbol{x})))$$
$$\mathcal{L}_\theta = \text{D}_{\text{KL}}[q_\phi(\boldsymbol{z} \,|\, \hat{\boldsymbol{x}}) \,\|\, \text{N}(\mathbf{0}, \mathbf{I})] + \lambda_\mathcal{Z} \, d_{\cos}(\boldsymbol{z}, \phi(\boldsymbol{\theta}(\mathbf{z}))) + \mathcal{L}_j, \tag{6}$$

where $\hat{\boldsymbol{x}}_{1:\frac{3}{4}M} \sim q_\theta(\boldsymbol{x} \,|\, \boldsymbol{z})$ with $\boldsymbol{z} \sim \text{N}(\mathbf{0}, \mathbf{I})$, and $\hat{\boldsymbol{x}}_{\frac{3}{4}M:M} \sim q_\theta(\boldsymbol{x} \,|\, \hat{\boldsymbol{z}}_{AB})$, with a set 3:4 ratio of regular and mixed samples of batch size $M$ and $j \sim \text{U}\{1, N\}$. Margin $M_{\text{gap}} = 0.5$ in low resolutions, then 0.2 for $128^2$ and 0.4 for $256^2$ (see Heljakka et al., 2019). To avoid discontinuities in $\boldsymbol{\alpha}$, we utilize a progressively-growing variation of $d_\rho$, where we first learn the $\boldsymbol{\alpha}$ in the lower resolutions (*e.g.*, 4×4). Each $\alpha_i$ corresponds to one pixel. Then, when switching to the higher resolution stage, we take take each parameter $\alpha_i$ that corresponds to pixels $p_{x,y}$ in the lower resolution, to initialize the $\alpha_j^{1 \times 4}$ that, in the higher resolution, corresponds to $p_{x,y}, p_{x+1,y}, p_{x,y+1}$ and $p_{x+1,y+1}$, respectively.

### 3.3   ENFORCING KNOWN INVARIANCES AT SPECIFIC LAYERS

The architecture and the cyclic training method also allow for a novel principled approach to leverage known scale-specific invariances in training data. Assume that images $x_1$ and $x_2$ share certain identical characteristics at some scales, but differ on others, with this information further encoded into $z_1$ and $z_2$, correspondingly. In the automodulator, we could expect to try to represent this information in decoder layers #$j$:$k$. Following the reasoning of the previous section, we then must have $\theta_{j:k}(\boldsymbol{\xi}^{j-1}, \boldsymbol{z}_1) = \theta_{j:k}(\boldsymbol{\xi}^{(j-1)}, \boldsymbol{z}_2)$. Let us assume that it is possible to represent the rest of the information in the images of that dataset in layers #1:$(j-1)$ and #$(k+1)$:$N$. This situation occurs, *e.g.*, when two images are known to differ only in high-frequency properties, fully representable in the 'fine' layers. Utilizing the independence of the layers, our goal is to have

$$\boldsymbol{\theta}_{k+1:N}(\boldsymbol{\theta}_{j:k}(\boldsymbol{\theta}_{1:j-1}(\boldsymbol{\xi}^{(0)}, \boldsymbol{z}_2), \boldsymbol{z}_1), \boldsymbol{z}_2) = \boldsymbol{\theta}_{k+1:N}(\boldsymbol{\theta}_{j:k}(\boldsymbol{\theta}_{1:j-1}(\boldsymbol{\xi}^{(0)}, \boldsymbol{z}_2), \boldsymbol{z}_2), \boldsymbol{z}_2)$$
$$= \boldsymbol{\theta}_{1:N}(\boldsymbol{\xi}^{(0)}, \boldsymbol{z}_2) = \boldsymbol{\theta}(\phi(\boldsymbol{x}_2)) \tag{7}$$

which turns into the optimization target (for some distance function $d$)

$$d(\boldsymbol{\theta}(\phi(\boldsymbol{x}_2)), \boldsymbol{\theta}_{k+1:N}(\boldsymbol{\theta}_{j:k}(\boldsymbol{\theta}_{1:j-1}(\boldsymbol{\xi}^{(0)}, \boldsymbol{z}_2), \boldsymbol{z}_1), \boldsymbol{z}_2)). \tag{8}$$

By construction of $\phi$ and $\boldsymbol{\theta}$, we immediately see this is equivalent to directly minimizing

$$\mathcal{L}_{\text{inv}} = d(\boldsymbol{x}_2, \boldsymbol{\theta}_{k+1:N}(\boldsymbol{\theta}_{j:k}(\boldsymbol{\theta}_{1:j-1}(\boldsymbol{\xi}^{(0)}, \boldsymbol{z}_2), \boldsymbol{z}_1), \boldsymbol{z}_2)) \tag{9}$$

whose complement term $\mathcal{L}'_{\text{inv}}$ you can construct by swapping $z_1$ with $z_2$ and $x_1$ with $x_2$. For each known invariant pair $\boldsymbol{x}_1$ and $\boldsymbol{x}_2$ of the minibatch, you can now add the terms $\mathcal{L}_{\text{inv}} + \mathcal{L}'_{\text{inv}}$ to $\mathcal{L}_\phi$ of Eq. (6). Note that in the case of $\boldsymbol{z}_1 = \boldsymbol{z}_2$, $\mathcal{L}_{\text{inv}}$ reduces to the regular sample reconstruction loss, revealing our formulation as a generalization thereof.

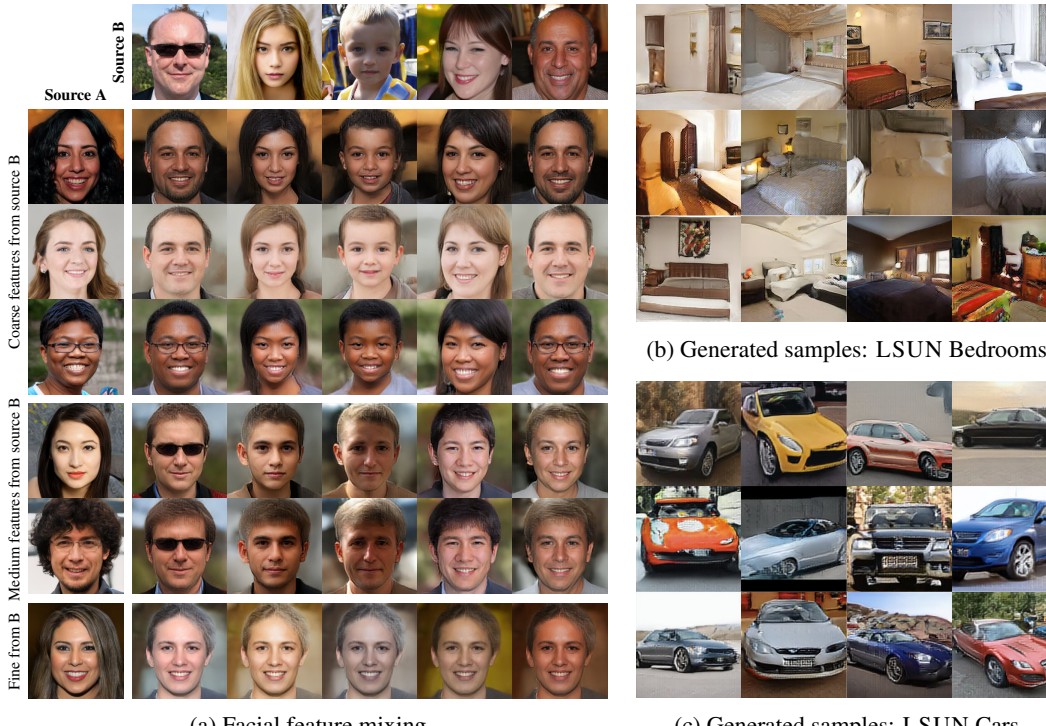

(a) Facial feature mixing

(b) Generated samples: LSUN Bedrooms

(c) Generated samples: LSUN Cars

Figure 4: (a) Style-mixing example using the same source images as Karras et al. (2019) (underlining that our model can directly work with real input images). (b–c) Random samples at $256\times256$.

As we push the invariant information to layers $\#j{:}k$, and the other information *away* from those layers, this reduces the number of layers available for the rest of the image information. Thus we may need to add extra layers to retain the overall decoder capacity. Note that in a pyramidal deconvolutional stack, the layers $\#j{:}k$ can safely span only one or two consecutive levels of detail.

## 4 EXPERIMENTS

We run our experiments on images using CELEBA-HQ (Karras et al., 2018), FFHQ (Karras et al., 2019), and LSUN Bedrooms and Cars (Yu et al., 2015). To quantify the image quality and diversity of random draws from the model at $256\times256$ resolution, we use the Fréchet inception distance (FID,

Table 1: Effect of loss terms, CelebA-HQ.

| Method (64×64, 20M steps) | FID-10k (normal) | FID-10k (mix) |
|---|---|---|
| Automodulator architecture | $\mathbf{11.54 \pm 0.32}$ | $15.90 \pm 0.22$ |
| + Style-mixing loss | $12.71 \pm 0.28$ | $14.88 \pm 0.16$ |
| + Robust loss (no L1 loss) | $12.23 \pm 0.21$ | $\mathbf{14.75 \pm 0.14}$ |

Heusel et al., 2017), which is comparable across models when sample size is fixed (Binkowski et al., 2018). However, FID is known to remain constant under changing precision-recall characteristics (Kynkäänniemi et al., 2019). We use LPIPS (Zhang et al., 2018) as the similarity metric, which has better correspondence to human evaluation than traditional L2 metrics. The degree of latent space disentanglement is often considered the most important property of a latent variable model. Qualitatively, it is the necessary condition for, *e.g.*, style mixing capabilities. Quantitatively, Karras et al. (2019) noted that, for a constant-length step in the latent space, a more entangled model will produce a larger overall perceptual change than a less entangled model. The extent of this change, with LPIPS as the perceptual space metric, is the basis of measuring disentanglement as Perceptual Path Length (PPL).

**Baseline Methods** For autoencoders, we compare to Balanced PIONEER by Heljakka et al. (2019), a vanilla VAE, and a more recent Wasserstein Autoencoder (WAE, Tolstikhin et al., 2017). For non-autoencoders, we compare to GLOW (Kingma & Dhariwal, 2018) and two recent GAN models: StyleGAN and Progressively Growing GAN (PGGAN, Karras et al., 2018).

Table 2: Performance in CELEBA-HQ (CAHQ), FFHQ, and LSUN Bedrooms and Cars. We measure LPIPS, Fréchet Inception Distance (FID), and perceptual path length (PPL). Resolution is $256{\times}256$, except *$128{\times}128$. For all numbers, **smaller is better**.

(a) Encoder–decoder comparison

| (Using CAHQ*) | LPIPS (cropped) | FID | PPL |
|---|---|---|---|
| B-PIONEER | **0.092** | **21.51** | 92.84 |
| WAE-AdaIn | 0.165 | 100.02 | 62.17 |
| WAE-classic | 0.162 | 108.93 | 236.82 |
| VAE-AdaIn | 0.267 | 114.52 | 83.52 |
| VAE-classic | 0.291 | 173.37 | 71.75 |
| Automodulator | 0.102 | 36.19 | **41.45** |

(b) Generative models comparison

| | FID (CAHQ) | FID (FFHQ) | FID (Bedrooms) | FID (Cars) | PPL (CAHQ*) | PPL (FFHQ) |
|---|---|---|---|---|---|---|
| StyleGAN | **5.17** | **4.40** | **2.65** | **3.27** | 50.08 | 195.9 |
| PGGAN | 7.79 | 8.04 | 8.34 | 8.36 | 81.33 | 412.0 |
| GLOW | 68.93 | — | — | — | 138.21 | — |
| B-PIONEER | 25.25 | — | 21.52 | 42.81 | 92.84 | — |
| Automodulator | 51.96 | 64.07 | 35.74 | 35.61 | **41.45** | 262.3 |

**Ablation** We illustrate the contribution of the style-mixing loss $\mathcal{L}_j$ and the robust loss $d_\rho$ on $64{\times}64$ CELEBA-HQ (Table 1). $\mathcal{L}_j$ increases the FID of regular random samples, but reduces it when two random latents are mixed. $d_\rho$ then reduces some of the FID of regular samples.

## 4.1 ENCODING, DECODING, AND RANDOM SAMPLING

In Table 2a, we compare encoder–decoder performance of the autoencoders on $128{\times}128$ CELEBA-HQ, with our proposed architecture ('AdaIn') and the corresponding regular architecture ('classic'). We measure LPIPS, FID and PPL. Our method has the best PPL, while Balanced PIONEER has the best FID (50k batch of generated samples compared to training samples).

Table 2b shows comparison of random sampling (examples in Figs. 4b–4c) performance via FID and latent space smoothness via PPL for $256{\times}256$ images on CELEBA-HQ, FFHQ and LSUN Cars and Bedrooms. The performance of the automodulator is comparable to the Balanced PIONEER on most data sets, but the FID is worse in general. Models that *only* generate random samples have clearly best FID results on all data sets (NB: a hyper-parameter search with various schemes was used in Karras et al., 2019, to achieve their PPL for FFHQ). We train on the actual 60k training set of FFHQ only (StyleGAN trained on all 70k images). We also evaluate the 4-way image interpolation capabilities in unseen FFHQ test images (Fig. 12 in the appendix) and observe smooth transitions. We emphasize that in GANs, such interpolations are often made between the codes of generated samples. As such, they cannot tell much about the recall characteristics of those models.

## 4.2 STYLE MIXING

We demonstrate the style-mixing capabilities of our model. For comparison with prior work, we use the source images from the StyleGAN paper (Karras et al., 2019). In Fig. 4a, we mix specific input faces so that the 'coarse' (latent resolutions $4{\times}4 - 8{\times}8$), 'intermediate' ($16{\times}16 - 32{\times}32$) or 'fine' ($64{\times}64 - 256{\times}256$) layers of the decoder use one input, and the rest of the layers use the other. Importantly, StyleGAN cannot take real inputs, so it can only mix between random images created by the model itself. For new input images, one must run a separate costly optimization process to determine the most fitting latent code. For our model, those images appear as completely new test images. Additional style mixing results are included in Figs. 13–14 in the appendix.

## 4.3 LEVERAGING INVARIANCES IN A WEAKLY SUPERVISED SETUP

We now consider the cases where there is knowledge of some scale-specific invariances within the training data. Here, we demonstrate a proof-of-concept experiment that uses the simplest image transformation possible: horizontal flipping. For the CELEBA-HQ face data, this provides us with pairs of images that share every other property except the azimuth rotation angle of the face, making the face identity invariant amongst each pair. Since the original rotation of faces in the set varies, the flip-augmented data set contains faces rotated across a wide continuum of angles. For further simplicity, we make an artificially strong hypothesis that the 2D projected face shape is the only relevant feature at $4{\times}4$ scale, and does not need to affect scales finer than $8{\times}8$. This lets us enforce the $\mathcal{L}_{\text{inv}}$ loss for layers #1–2. Since we do not want to restrict the scale $8{\times}8$ for the shape features alone, we add an extra $8{\times}8$ layer after layer #2 of the regular stack, so that layers #2–3 both operate at $8{\times}8$, layer #4 only at $16{\times}16$, *etc.* Now, with $z_2$ that corresponds to the horizontally flipped counterpart of $z_1$, we have $\boldsymbol{\theta}_{3:N}(\boldsymbol{\xi}^{(2)}, z_1) = \boldsymbol{\theta}_{3:N}(\boldsymbol{\xi}^{(2)}, z_2)$. Our choices amount to $j = 3, k = N$,

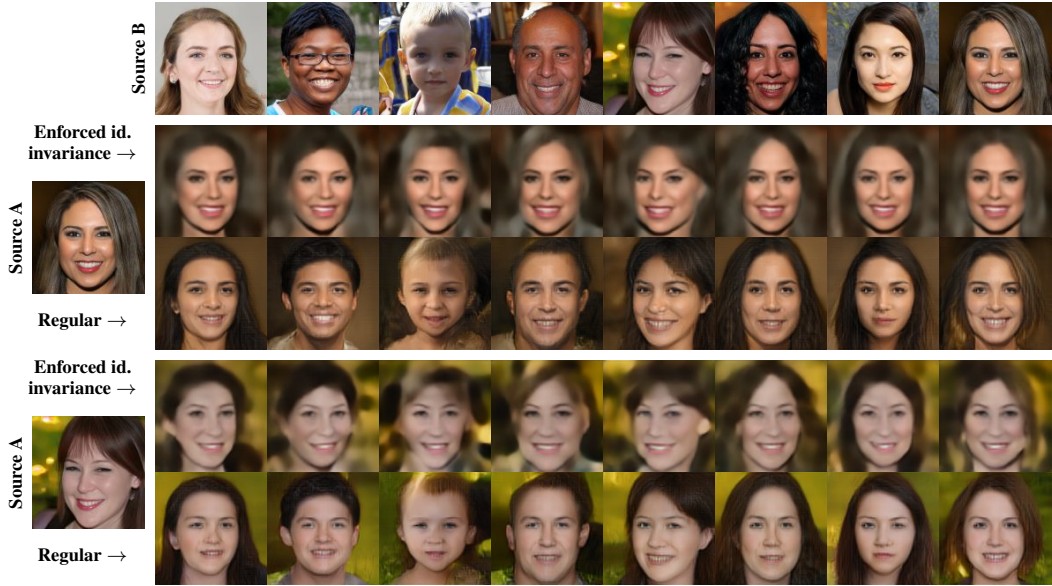

Figure 5: The effect of training with face identity invariance enforcement under azimuth rotation. We generate images with the 'non-coarse' styles from the source images (left), and the 'coarse' styles from each top row image, in turn. With 'Enforced id. invariance', the top row only drives the face pose while conserving identity. In comparison, the 'Regular' training lets the top row also affect other characteristics, including identity. Here, the actual data that defined the invariance was too simple to allow learning fine-grained control of the hair at coarse levels, making it blurry.

allowing us to drop the outermost part of Eq. (9). Hence, our additional encoder loss terms are

$$\mathcal{L}_{\mathrm{inv}} = d(\boldsymbol{x}_2, \boldsymbol{\theta}_{3:N}(\boldsymbol{\theta}_{1:2}(\boldsymbol{\xi}^{(0)}, \boldsymbol{z}_2), \boldsymbol{z}_1)) \quad \text{and} \tag{10}$$

$$\mathcal{L}'_{\mathrm{inv}} = d(\boldsymbol{x}_1, \boldsymbol{\theta}_{3:N}(\boldsymbol{\theta}_{1:2}(\boldsymbol{\xi}^{(0)}, \boldsymbol{z}_1), \boldsymbol{z}_2)). \tag{11}$$

Fig. 5 shows the results after training with the new loss (50% of the training samples flipped in each minibatch). With the invariance enforcement, the model forces decoder layers #1–2 to only affect the pose. We generate images by driving those layers with faces at different poses, while modulating the rest of the layers with the face whose identity we seek to conserve. We receive variations of the original face that only differ in terms of the pose, unlike in regular automodulator training.

## 5 DISCUSSION AND CONCLUSION

In this paper, we proposed the first generative autoencoder model with a hierarchical latent representation that supports controllable image generation and editing, conditional image sampling by fixing styles of specific layers, and style-mixing of real images. In our model, the latent vector independently modulates each decoder layer. The model outperforms other generative autoencoders in terms of latent space disentanglement and matches them in faithfulness of reconstructions, with slight reduction of output sample quality. We use the term *automodulator* to denote any autoencoder that uses the latent code only to modulate the statistical properties of the information that flows through the layers of the decoder. Such decoders could also include, *e.g.*, 3D or graph convolutions.

Potential future applications include introducing completely independent 'plugin' layers or modules in the decoder, trained afterwards on top of the pretrained base automodulator, leveraging the mutual independence of the layers. The affine maps themselves could also be re-used across domains, potentially offering mixing of different domains. Such examples highlight that the range of applications of our model is far wider than the initial ones shown here, making the family of automodulators a viable alternative to state-of-the-art autoencoders and GANs. Upon acceptance for publication, our source code will be released at http://github.com/anonymized.

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

# A    APPENDIX

In the appendix, we include further details on training and complement the results in the main paper with examples of random samples, reconstruction, latent space interpolations, and style mixing.

## A.1    TRAINING DETAILS

The training method follows largely those in Heljakka et al. (2019), with progressively growing symmetric encoder and decoder, and decreasing the batch size when moving to higher resolutions. The encoder and decoder consist of 7 blocks each containting two residual blocks with a $3\times3$ filter. In the encoder, these are followed by a spectral normalization operation (Miyato et al., 2018) and binomial filtering. In the decoder, by AdaIn normalization and binomial filtering. A leaky ReLU ($p = 0.2$) is used as activation. In the encoder, each block halves the resolution of the convolution map, while in the decoder, each block doubles it. The output of the final encoder layers is flattened into a 512-dimensional latent block. As in Karras et al. (2019), the block is mapped by affine mapping layers so that each convolutional layer $C$ in the decoder block is preceded by its own fully connected layer that maps the latent to two vectors each of length $N$, when $N$ equals the number of channels in $C$.

Each resolution phase until $32\times32$ for all data sets use a learning rate $\alpha = 0.0005$ and thereafter 0.001. Optimization is done with ADAM ($\beta_1 = 0, \beta_2 = 0.99, \epsilon = 10^{-8}$). After the first two resolution steps, the KL margin is turned on and fixed to 0.5. The length of training phases amounts to 2.4M training samples until $64\times64$ resolution phase, which lasts for 10.4M samples. For FFHQ and CelebAHQ, the $128\times128$ phase uses 13.0M samples while LSUN Bedrooms and Cars use 10.0M samples. The final $256\times256$ phase uses 7–10M samples for each data set. The training of the final stage was generally cut off when reasonable FID results had been obtained. More training and learning rate optimization would likely improve results. With two Titan V100 GPUs for pre-training stages and four GPUs for the $256\times256$ stage, the training time for CELEBA-HQ and FFHQ were 18 days each, and for LSUN Bedfrooms 17 days and Cars 19 days.

For evaluating the model after training, a moving expontential running average of generator weights was used, as in both Karras et al. (2018) and Heljakka et al. (2019). For all data sets, training/test set splits were used as given or defined by data set authors, except for LSUN Cars, where we used 4,968,695 samples for training and 552,061 for testing. Note that in regular GAN training, complete data sets are often used without train/test split, leading them to use larger training sets.

For baselines, we used pre-trained models for StyleGAN, PGGAN, PIONEER, and GLOW with default settings provided by the authors. We trained the VAE and WAE models manually. For all VAE baselines the weight for KLD loss was 0.005. For all WAE baseline, we used the WAE-MMD algorithm. The weights for the MMD loss with automodular architecture (WAE-AdaIn) was 4 and with Balanced PIONEER (WAE-classic) architecure it was 2. For VAEs, the learning rate for the encoder was 0.0001, and for the generator 0.0005. For WAEs, the learning rate for both was 0.0002.

For evaluating the encoding and decoding performance, we used 10k unseen test images from the FFHQ data set, cropped the input and reconstruction to $128\times128$ as in Karras et al. (2019) and evaluated the LPIPS distance between the inputs and reconstructions. We evaluated 50k random samples in all data sets and compare against the provided training set. The GLOW model has not been shown to work with $256\times256$ resolution on LSUN (the authors show qualitative result only for $128\times128$). Training of PIONEER did not converge on FFHQ, however we believe this is an issue with the default hyper-parameters not suitable for FFHQ.

For Perceptual Path Length (PPL), we repeatedly take a random vector of length $\varepsilon = 10^{-4}$ in the latent space, generate images at its endpoints, crop them around mid-face to $128\times128$ or $64\times64$, and measure the LPIPS between them (Karras et al., 2019). PPL equals the scaled expectation of this value (for a sample of 100k vectors).

Pre-trained models for PGGAN and GLOW were used with default settings provided by the authors. Note that we train on the actual 60k training images of FFHQ only (unlike StyleGAN that trained on all 70k images).

## A.2 RANDOM SAMPLES

Our model is capable of fully random sampling by specifying $z \sim \mathrm{N}(\mathbf{0}, \mathbf{I})$ to be draws from a unit Gaussian. Fig. 6–8 show samples from an automodulator trained with the FFHQ/CELEBA-HQ/LSUN data sets up to resolution $256 \times 256$. The samples here indicate the full range of samples and face features the model can support.

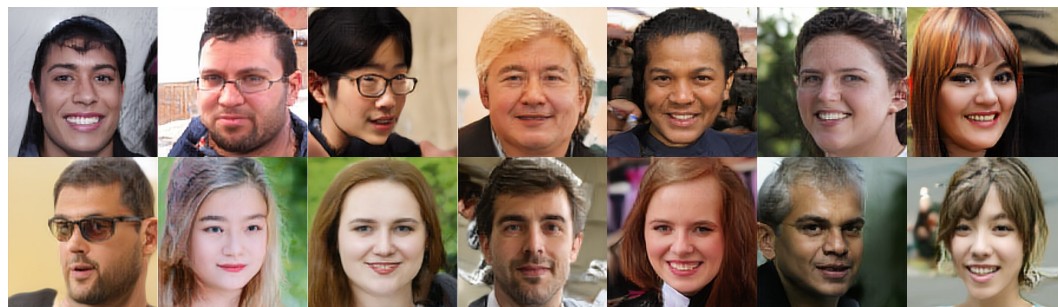

Figure 6: Random samples from the automodulator trained on FFHQ at a resolution $256 \times 256$.

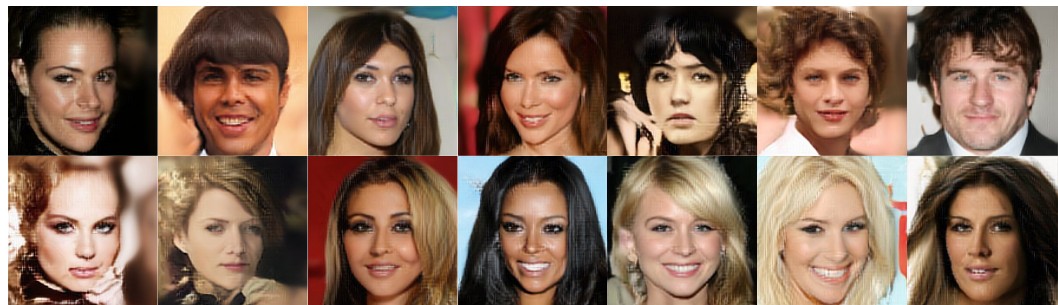

Figure 7: Random samples for an automodulator trained on CELEBA-HQ at resolution $256 \times 256$.

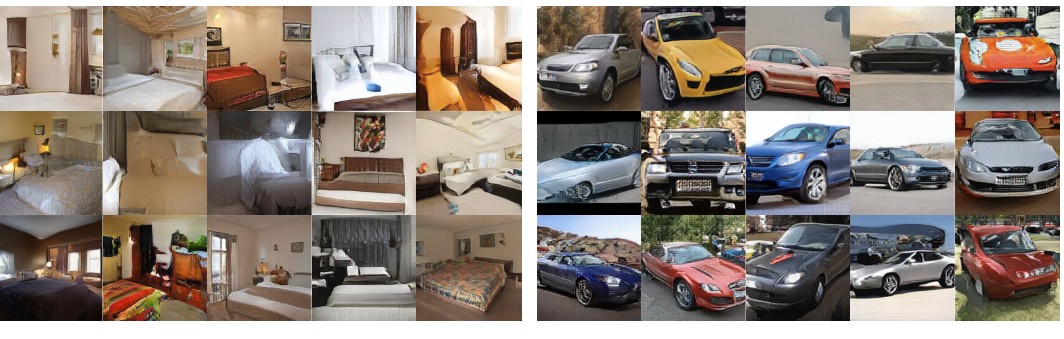

(a) LSUN Bedrooms        (b) LSUN Cars

Figure 8: Additional samples from an automodulator trained on LSUN Bedrooms and Cars a resolution of at $256 \times 256$.

## A.3 RECONSTRUCTIONS

We include examples of the reconstruction capabilities of the automodulator at $256 \times 256$ in for uncurated test set samples from the FFHQ and CELEBA-HQ data sets. These examples are provided in Figs. 9–10.

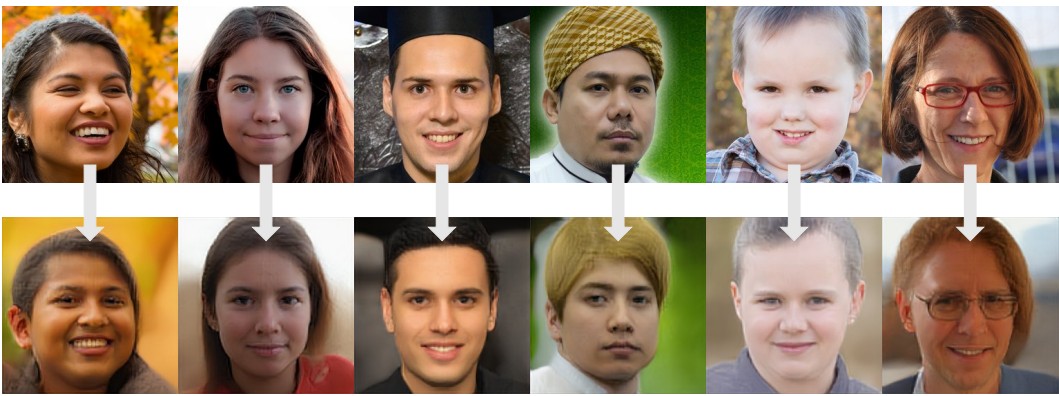

Figure 9: Uncurated examples of reconstruction quality in 256×256 resolution with images from the FFHQ test set (top row).

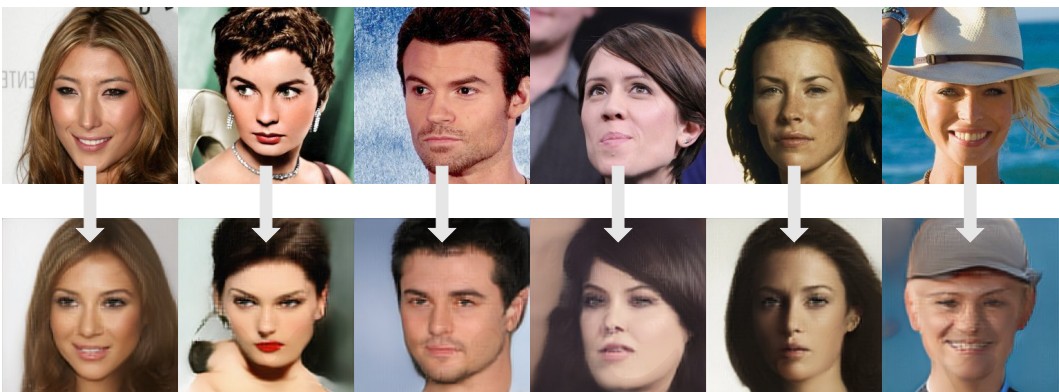

Figure 10: Uncurated examples of reconstruction quality in 256×256 resolution with images from the CELEBA-HQ test set (top row).

## A.4 CONDITIONAL SAMPLING

The automodulator directly allows for conditional sampling in the sense of fixing a latent encoding $z_A$, but allowing some of the modulations come from a random encoding $z_B \sim N(\mathbf{0}, \mathbf{I})$. In Fig. 11, we show conditional sampling of 128×128 random face images based on 'coarse' (latent resolutions $4\times4 - 8\times8$) and 'intermediate' ($16\times16 - 32\times32$) latent features of the fixed input. The input image controls the coarse features (such as head shape, pose, gender) on the top and more fine features (expressions, accessories, eyebrows) on the bottom.

## A.5 STYLE MIXING AND INTERPOLATION

The well disentangled latent space allows for interpolations between encoded images. We show regular latent space interpolations between the reconstructions of new input images (Fig. 12).

As two more systematic style mixing examples, we include style mixing results based on both FFHQ and LSUN Cars. The source images are unseen real test images, not self-generated images. In Figs. 13 and 14 we show a matrix of cross-mixing either 'coarse' (latent resolutions $4\times4 - 8\times8$) or 'intermediate' ($16\times16 - 32\times32$) latent features. Mixing coarse features results in large-scale changes, such as pose, while the intermediate features drive finer details, such as color.

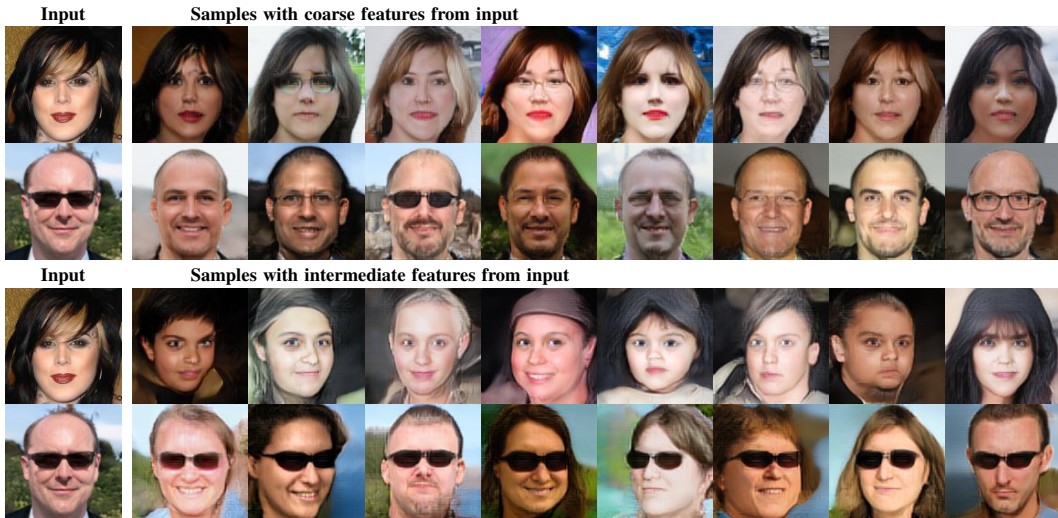

Figure 11: Conditional sampling of 128×128 random face images based on 'coarse' (latent resolutions 4×4 − 8×8) and 'intermediate' (16×16 − 32×32) latent features of the fixed input. The input image controls the coarse features (such as head shape, pose, gender) on the top and more fine features (expressions, accessories, eyebrows) on the bottom.

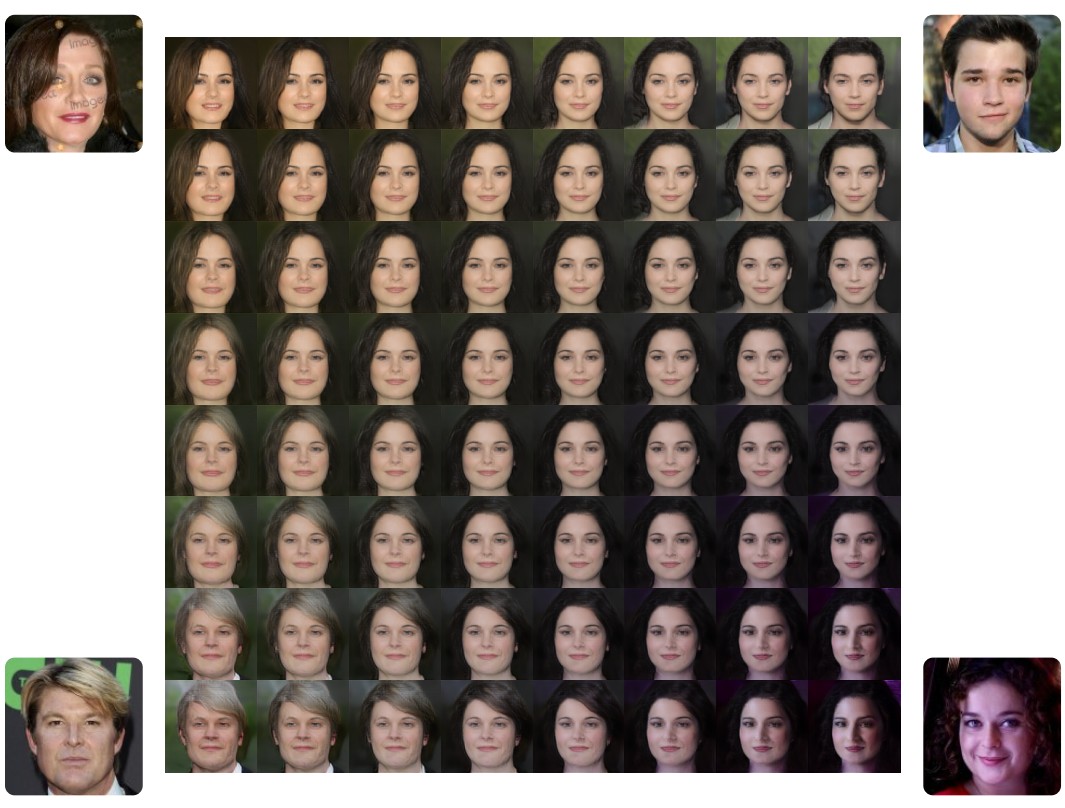

Figure 12: Interpolation between random test set CELEBA-HQ images in 128×128 (in the corners) which the model has not seen during training. The model captures most of the salient features in the reconstructions and produces smooth interpolations at all points in the traversed space.

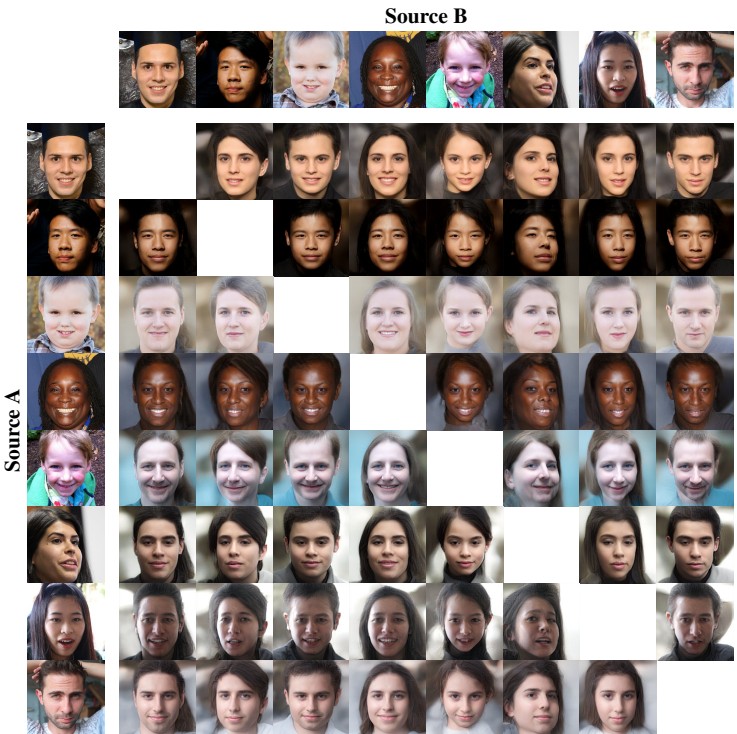

(a) Using 'coarse' (latent resolutions $4{\times}4 - 8{\times}8$) latent features from B and the rest from A.

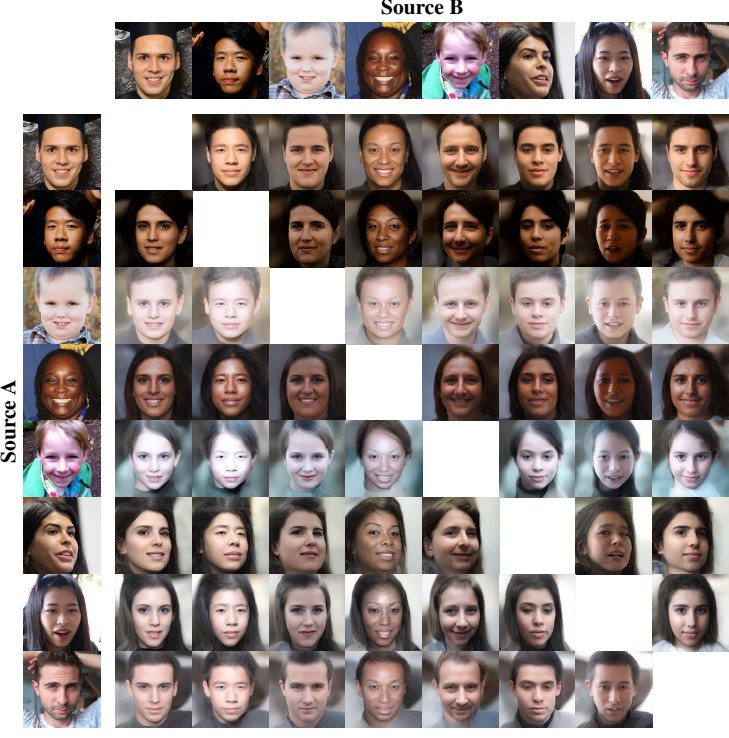

(b) Using the 'intermediate' ($16{\times}16 - 32{\times}32$) latent features from B and the rest from A.

Figure 13: Style mixing of FFHQ face images. The source images are unseen real test images, not self-generated images.

**Source B**

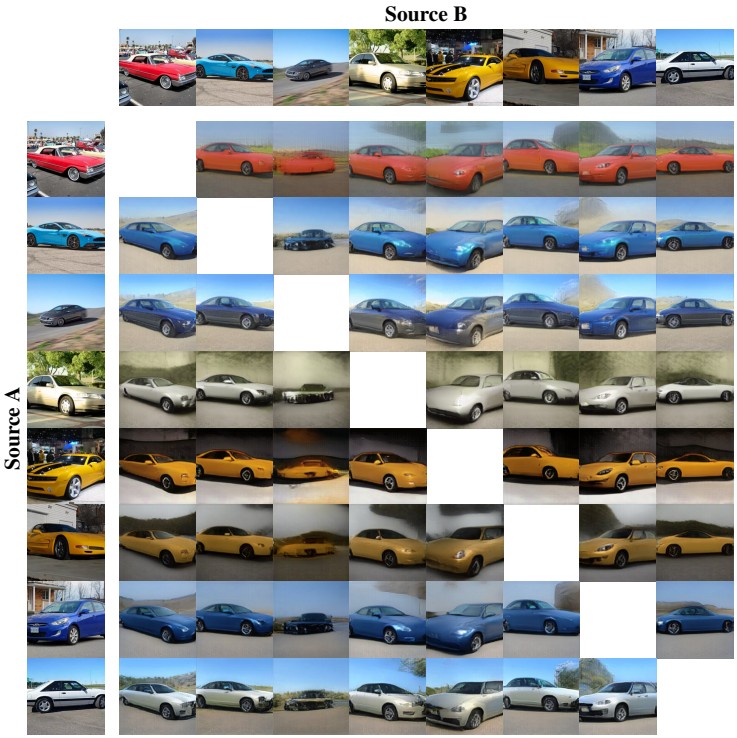

(a) Using 'coarse' (latent resolutions 4×4 – 8×8) latent features from B and the rest from A. Most notably, the B cars drive the car pose.

**Source B**

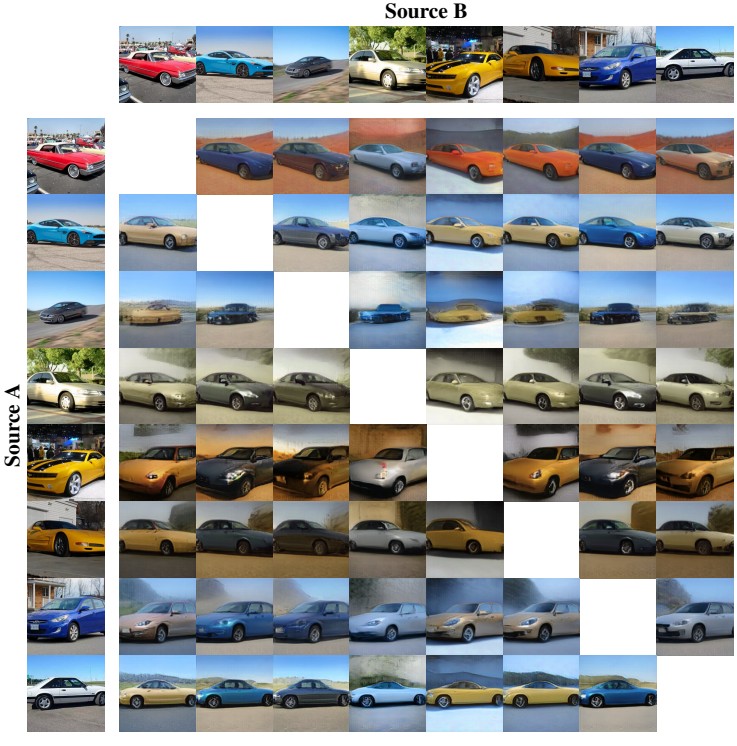

(b) Using the 'intermediate' (16×16 – 32×32) latent features from B and the rest from A.

Figure 14: Style mixing of LSUN Cars. The source images are unseen real test images, not self-generated images.

