# OpenReview forum: "Deep automodulators"
_ICLR.cc/2020/Conference — Reject_

### Official Review · AnonReviewer4 · 2019-10-23
**Official Blind Review #4**

**Rating:** 3

**Review:**

The submission proposes an autoencoder architecture which combines two recent GAN-based architectural innovations, namely the progressive growing of the decoder architecture (as well as the encoder architecture in this case) and the use of the encoded representation to modulate the decoder via a feature-wise transformation mechanism.

I think the overall idea behind the paper is valuable, but I don’t think the submission meets the acceptance bar from a clarity point of view. I also have concerns with its characterization of the literature.

Clarity-related comments:

- “This allows the layers to work independently of each other.” This is an imprecise use of the term “independently”. How can two layers work independently if one’s input is the output of the other?
- “The reconstructed samples could be re-introduced to the encoder, repeating the process, and requiring consistency between passes.” The rest of the paragraph is built on the premise that this is a desirable property, but I’m not sure I understand why this is a desirable property in the first place.
- How to define “disentanglement” in the context of representation learning is in itself an unsettled question as far as I’m aware, but the submission uses the term without an intuitive or formal definition. What do the authors mean by “disentangled representation”? What is measured by perceptual path length (PPL), and in which ways does PPL relate to the author’s definition of “disentangled representation”?
- “[...] a new autoencoder-like model with powerful properties not found in regular autoencoders, including style transfer.” The term “style transfer” is overloaded; what do the authors mean?
- The submission defines AdaIn as a way to combine “content” and “style”, and defines the style “y” in terms of mean and variance. In the context of the AdaIn paper, this makes sense: the instance normalization shifting and scaling coefficients are heuristically defined as the channel-wise means and standard deviations of a “style” stack of feature maps. However, in the context of this submission I’m not sure this definition makes as much sense: the instance normalization shifting and scaling coefficients are the result of a linear projection of the latent representation and do not involve the channel-wise means and standard deviations of an external stack of feature maps; is this correct?
- “This setup follows the same logic as that of Karras et al. (2019), but we do not require an ad-hoc disentanglement stack.” Can the authors clarify what they mean by an “ad-hoc disentanglement stack”?
- Section 3.2 uses some notation for the encoder without introducing it first. I believe the only way to understand that \phi(x) refers to the encoder network is to look at Figure 2.
- Section 3.2 as a whole is hard to follow, in part due to the use of imprecise language (“mutually independent”, “representation of those levels disentangled in z”). At some point probability distributions are introduced (up until now the reader is operating under the assumption that the model is an autoencoder with no probabilistic interpretation), and mutual information is mentioned to justify an L2 reconstruction loss in z-space (which I would argue is an instance of mathiness that does not serve the reader’s comprehension). Can the authors explain in plain language how layer-specific losses are defined and how the complete loss is obtained?
- The submission presents model samples, but as far as I can tell the procedure for sampling is not provided. Unlike VAEs, autoencoders do not explicitly model the empirical distribution -- although reconstruction in denoising autoencoders is related to the score of the empirical distribution (Alain & Bengio, 2014). How are samples obtained from the trained model?

Literature-related comments:

- The use of normalization layers to implement feature-wise transformation mechanisms is fairly widespread nowadays, but for instance normalization specifically the work of Dumoulin et al. (2017) pre-dates that of Huang & Belongie (2017). Both are cited by Karras et al. (2019) in relation to AdaIn (which is termed “conditional instance normalization” in Dumoulin et al. (2017)).
- I disagree with the characterization of ALI/BiGAN as “hybrid models that combine the properties of VAEs and GANs”: unlike AAE and AVB, which minimize KL-divergence terms in the VAE loss adversarially, the objective for ALI/BiGAN is purely adversarial. I would also include IAE (Makhzani et al., 2018) and BigBiGAN (Donahue et al., 2019) in the list of GAN variants that incorporate an inference mechanism.
- The submission repeatedly asserts that GANs lack an inference mechanism: “Unlike GANs, autoencoder models can directly operate on input samples.”; “To work on new input images, GANs either need to be extended with a separate encoder, or inverted [...].”; “[...] GANs show good image quality, but have no built-in encoding mechanism [...]”; “[...] the problem with GANs is that they lack the encoder [...]”. This is false: see for example ALI, BiGAN, and BigBiGAN. The problem in my opinion is elsewhere: the kinds of reconstructions these models yield are not suited to the downstream applications investigated in this submission, because they oftentimes fail to preserve low-level details.

References:

- Alain, G., & Bengio, Y. (2014). What regularized auto-encoders learn from the data-generating distribution. The Journal of Machine Learning Research, 15(1), 3563-3593.
- Dumoulin, V., Shlens, J., & Kudlur, M. (2017). A learned representation for artistic style. In Proceedings of the International Conference on Learning Representations.
- Makhzani, A. (2018). Implicit autoencoders. arXiv:1805.09804.
- Donahue, J., & Simonyan, K. (2019). Large scale adversarial representation learning. arXiv:1907.02544.

**Experience Assessment:**

I have published in this field for several years.

**Review Assessment: Checking Correctness Of Derivations And Theory:**

N/A

**Review Assessment: Checking Correctness Of Experiments:**

I assessed the sensibility of the experiments.

**Review Assessment: Thoroughness In Paper Reading:**

I read the paper thoroughly.

---

> ### Author Response · Authors · 2019-11-15
> **Response to Review #4**
>
> Thank you for the detailed requests and literature pointers. We hope that our extensive rewrites of Sec. 3 especially will address most of these questions. We also added the references to the other models you provided.
>
> The parts about layer independence and style transfer have been rephrased. A working definition for disentanglement is now (albeit very shortly) given in Sec. 1, explanation of the related PPL added in Sec. 4, and some details of how we calculated it were added in the Appendix.
>
> As for the use of style transfer and AdaIn in our context, please find the updated description in Sec. 3.1. which hopefully makes it clearer that indeed, also in this context, the shifting and scaling coefficients relate to the feature maps, in essentially the same way as they do in StyleGAN (Karras, 2019), and hence allow using the (overloaded) term ‘style’.
>
> We removed the term "ad hoc disentanglement stack" and replaced it with the original term "mapping layer", and moved it to Related Work section.
>
> As for $\phi(x)$, it was actually mentioned in the very beginning of 3.1. But nonetheless, we have now added more background information from Ulyanov (2018) and Heljakka (2018).
>
> As for the layer-specific loss, it can be summarized like this: we generate a random latent z1, and start driving the decoder with it, layer by layer, until we reach some layer J. There, we record the intermediate result of the decoding, and then pick another latent z2, and continue driving the decoder with that one, instead. Once we have decoded the full image, we encode it again into z12, and start decoding again, until we reach again layer J. At that point, we consider the reconstruction loss between the previous intermediate decoding result of z1 and the new corresponding result of z12, thus trying to make the model ignore the effect of z2 until we move again onwards from layer J.
>
> As for the probabilistic interpretation and random sampling in AGE models, we added more explanation of both. AGE latent space is, like regular VAE's, a 512-dimensional unit hypersphere. We can sample from it exactly as we do from VAE latent space. This sampling is also utilized at every training step.
>
> Your point about the mis-characterization of ALI/BiGAN as VAE-GAN hybrids, from the point-of-view of their loss objective, is correct. We corrected our wording in that regard.
>
> We have made the wording in our referring to GANs and GAN inference mechanisms more precise (if not perfect), in alignment with your points. However, we’d be inclined to say that the terminology is somewhat fuzzy here, in terms of whether a GAN with an encoder should be called "GAN" or "GAN with a separate encoder". Consider, for instance, this use of terminology in the ALI paper that you refer to (https://openreview.net/pdf?id=B1ElR4cgg): "However, GANs lack an efficient inference mechanism", "Our approach...casts the learning...in an GAN-like adversarial framework", "ALI bears close resemblance to GAN, but it differs from it in the two following ways", etc. If you can recommend an authoritative source for what are the necessary and sufficient conditions for being a GAN, we would be happy to adapt to it!

---

### Official Review · AnonReviewer3 · 2019-10-29
**Official Blind Review #3**

**Rating:** 3

**Review:**

Deep Automodulators introduces a generative autoencoder architecture that replaces the canonical encoder decoder autoencoder architecture with one inspired by StyleGAN. The encoder interacts with the decoder by modulating layer statistics via Adaptive Instance Normalization (AdaIN) conditioned on the latent. The paper trains this architecture with the loss framework of the Adversarial Generator–Encoder (AGE) and utilizes the progressive growing trick originally introduced in Progressive GAN which is also adapted by the Pioneer models, recent followups to AGE.

The use of AdaIN conditioning across multiple layers and multiple scales (like StyleGAN) and the ability to directly compute latent codes via the encoder allows the authors introduce a disentanglement objective L_j and also an invariance objective L_inv to help encourage these properties in the models via consistency objectives

The paper shows results demonstrating StyleGAN style coarse/fine visual transfer on two high quality face datasets (importantly this is demonstrated on real inputs rather than samples as in StyleGAN) as well as respectable sample quality on LSUN Bedrooms and the LSUN Cars dataset.

My decision is weak reject. Overall, I think the paper is promising and shows a nice combination of efficient latent inference and controllable generation but the authors do not include ablations to validate some of their core contributions such as the L_j objective. Additionally, the improved controllability of the approach seems to unfortunately result in lower reconstruction quality than direct prior work such as Balanced Pioneer and this potential tradeoff is not investigated/discussed.

To expand a bit, there are three changes from that prior work that that stood out to me. 1) The StyleGAN inspired architecture 2) the disentangling objective L_j and 3) using the loss function dρ of Barron 2019. Successful ablations to demonstrate the importance of 2) to the presented results as well as better motivating / demonstrating the impact of including 3) would raise my score to an weak acceptance.

My other concern is that the reconstruction quality seems noticeably lower than that of the proceeding work, Balanced Pioneer. This is reflected in its 10% reduction in LPIPS compared to the Automodulator’s paper. In general there also seems to be noticeable grid artifacts in the samples across all datasets samples/reconstructions, which don’t seem as prominent in Balanced Pioneer. It is not immediately clear why this is the case and additional investigation of this, such as checking whether this is due to the introduction of the disentanglement objective, or the inclusion of the Barron 2019 loss function would be informative.

Additional Comments:

Each subsection of 3 could be improved by providing a brief introduction to the motivation for and aim of each contribution before launching directly into how it is implemented / achieved. Without that bit of context on the goals of each subsection, it was more difficult to follow along with what was being done and why.

The presentation of L_j with lots of inlined equations intermixed with text gets a bit difficult to read / follow.

**Experience Assessment:**

I have published in this field for several years.

**Review Assessment: Checking Correctness Of Derivations And Theory:**

N/A

**Review Assessment: Checking Correctness Of Experiments:**

I assessed the sensibility of the experiments.

**Review Assessment: Thoroughness In Paper Reading:**

I read the paper at least twice and used my best judgement in assessing the paper.

---

> ### Author Response · Authors · 2019-11-15
> **Response to Review #3**
>
> Thank you for the positive comments and encouragement for improvements. We have now added an ablation study that specifically looks into the effect of L_j and loss term d_rho from (Barron 2019). We don't think that either of them are absolutely critical, but Lj clearly improves style-mixing sample quality. As you see in the new Table 1, when measured within the constant budget of training steps, Lj essentially trades general random sampling performance for improved style-mix sampling performance. Note that random sampling with mixed samples (i.e. producing each random sample from 2+ different latents) is a more direct way to measure the disentanglement of scale-specific properties than PPL, but we can only use it when comparing across models that can do such mixing. (Measuring LPIPS/PPL differences reliably would require repeating the comparisons in 128x128 or higher.) We would not necessarily consider the use of d_rho an important contribution of the paper, but it seems to improve results slightly.
>
> The reduced reconstruction quality and grid artifacts, in comparison to Balanced Pioneer, seem to largely go away after fixing the implementation issue (see item #3 in our "Updated Version" comment, and esp. the updated Fig. 6 and 9).
>
> We have improved the subsections of Sec. 3 in many ways, we hope you will find the improvements significant.

---

### Official Review · AnonReviewer1 · 2019-10-30
**Official Blind Review #1**

**Rating:** 6

**Review:**

The paper makes the following contribution:
- using the AdaIn architecture proposed by Karras et al., 2019 with the autoencoding architecture of AGE/PIONEER;
- a cyclic loss to enforce disentangling between different layers;
- a method to enforce invariances at specific layers.

The adaptation of the AdaIn architecture in an autoencoding fashion (a la AGE/PIONEER) is sensible and well motivated, combining state-of-the-art generator while allowing inference in a compact setting (i.e. not requiring an additional discriminator).

The other contribution are harder to read and the writing should be improved.
The cyclic loss should be better described. The notation of the KL divergence is confusing if you are using the KL divergence defined by AGE/PIONEER and will need to be explained. I will also assume that d_cos is the cosine loss as defined by the PIONEER paper. This should be mentioned as well.
The method to enforce invariance is also not clear to me. While the authors introduce F as a "known invariance", it is unclear what role it plays in the cost function. Is F an invariant on which we measure this reconstruction loss d? What is d? Explaining that might shed light on the result Figure 5, e.g. why the images become blurry when doing this rotation.

The experiment demonstrates the sampling quality of the model and the transfer of features at different level (coarse-medium-fine) Figure 4. It is unclear what was the contribution of the layer specific loss metric to allow that feature transfer. It seems from Figure 5 the invariance objective has been roughly satisfied but at the cost of a significant drop in image quality.

The clearest contribution from this paper is definitely the AGE/PIONEER approach to train the AdaIn architecture. The two other contributions are unclear, both in their explanation and in what they contribute: the layer specific loss not compared to an architecture just trained in an AGE way, and the enforcing of invariance, although filling its objective, might deteriorate other desirable properties of the model (e.g. sample quality).

**Experience Assessment:**

I do not know much about this area.

**Review Assessment: Checking Correctness Of Derivations And Theory:**

N/A

**Review Assessment: Checking Correctness Of Experiments:**

I assessed the sensibility of the experiments.

**Review Assessment: Thoroughness In Paper Reading:**

I read the paper at least twice and used my best judgement in assessing the paper.

---

> ### Author Response · Authors · 2019-11-15
> **Response to Review #1**
>
> Thank you for the positive assessment and several appropriate remarks. In the updated draft, we have now made the relationship to AGE/PIONEER objectives clearer, addressing your points about KL divergence and d_cos, which certainly improves the readability of the paper. Understandably, detailed explanation of the original AGE equations would not have fit the paper, but we did our best.
>
> You are absolutely right about the lack of clarity in explanation of cyclic loss and invariance enforcement. We did considerable improvements to the text in those sections. We removed the 'F' notation in the invariance section since it seemed confusing to us, too. The invariant part that F notation was used for is captured in the decoder layers, where the key idea is that it takes the form of a reconstruction loss in the image space, but such that the reconstruction of input sample x1 has been affected by another training sample x2, but only at those exact decoder layers that should treat both samples identically. We made this also more explicit in Eqs. (9–11).
>
> The effect of cyclic loss has been investigated in the new ablation study (up to 64x64 resolution).
>
> For Fig. 5, you are correct in that in the presented experiment, the invariance objective reduces image quality in terms of making the images more blurry. We believe this is simply due to lack of capacity and crude "horizontal flip" training data. The experiment using the L_inv (invariance loss) was intended just to introduce the idea; a complete paper (or several) could be written to explore it further with more emphasis on output quality. As you say, the experiment presently fulfils the objective we claim: That the identity/pose distinction is clearly induced into the model by the training. Just to be clear, L_inv is only used in Fig. 5, and none of the main experiments use it.

---

### Author Response · Authors · 2019-11-15
**Updated Version**

We thank all the reviewers for their encouraging feedback and helpful comments, especially for pointing out parts that needed clarification, and the detailed advice on literature references and terminology by R4. We have addressed all the concerns raised by the reviewers, with detailed replies provided below.

The revised version (as of 2019-11-15) incorporates the following improvements:
1. TEXT: As all reviewers correctly remarked that clarity was lacking in some sections, we have made considerable improvements in the text, even rewriting complete paragraphs for better exposition.
2. ABLATION: As R1 and R3 pointed out, the paper benefits from an ablation study to disseminate the contributions of the loss function components. Such an ablation study was carried out and has now been added to Sec. 4.
3. IMAGE QUALITY FIX: As R3 noted, there were visible grid artifacts in some images and some weakness in some of the metrics. Right before the Discussion Period, we found a bug in our implementation of the encoder. After fixing this bug and slightly re-adjusting the KL margin term, we re-ran the FFHQ experiments, and the grid artifacts reduced considerably. We have updated most of the affected FFHQ images (Fig. 4a, 6, 9 and 13) and the FFHQ metrics in Table 2. For the final revision, we will also re-run the CelebA-HQ and the rest of the experiments that could be similarly affected, and expect to improve the rest of the images similarly. Because none of the reviewers considered the performance to be a bottleneck issue in the first place, we consider the probable forthcoming improvements only a bonus in this case.

Detailed list of changes:
• Sec. 1–2: Mostly incorporated improvements advised by R4, including some additional references and more precise framing of GANs vs. "GANs with encoder" (we did our best – there is no 100% consensus on the terminology).
• Sec. 3.1: Improved the clarity of explanation involving AdaIn (related to questions by R4).
• Sec. 3.2: Rewrote parts of the Layer-specific loss (Lj) to improve clarity. This includes making the previously inlined key equations formatted on their own lines.
• Sec. 3.2: For more context (covering various points raised by R1 and R4), we have included more exposition of the original AGE and PIONEER loss function and explanation of each loss term, as well as our departure of the original formulation, and described more clearly the nuances of deterministic inference and random sampling during training and evaluation.
• Sec. 3.2–3.3: Switched the representation of some terms to more intuitive forms and made e.g. variable indices between equations more consistent
• Sec. 3.3: Rewrote this short section for more clarity.
• Sec. 4: Added the ablation study (requested by R1 and  R3) and explanation of PPL (requested by R4).
• Sec. 4.3: Improved explanation of the "Invariances" experiment (requested by R3), as well as fixed a mistake in the description of how exactly the equations were used.
• Changes in other parts were necessary mostly to compress the paper, since the reviewers made several (well justified) requests for more thorough explanations.

---

### Decision · Program_Chairs · 2019-12-19

**Decision:**

Reject

**Comment:**

The manuscript proposes an autoencoder architecture incorporating two recent architectural innovations from the GAN literature (progressive growing & feature-wise modulation), trained with the adversarial generator-encoder paradigm with a novel cyclic loss meant to encourage disentangling, and procedure for enforcing layerwise invariances. The authors demonstrate coarse/fine visual transfer on generative modeling of face images, as well as generative modeling results on several Large Scale Scene Understanding (LSUN) datasets.

Reviewers generally found the results somewhat compelling and the ideas valuable and well-motivated, but criticized the presentation clarity, lack of ablation studies, and that the claims made were not sufficiently supported by the empirical evidence. The authors revised, and while it was agreed that clarity was improved, some reviewers were still not satisfied with the level of clarity (the revision appeared at the very end of the discussion period, unfortunately not allowing for any further refinement). Ablation studies were added in the revised manuscript, which were appreciated, but seemed to suggest that the proposed loss function was of mixed utility: while style-mixing quantitatively improved, overall sample quality appeared to suffer.

As the reviewers remain unconvinced as to the significance of the contribution and the clarity of its presentation, I recommend rejection at this time, while encouraging the authors to further refine the presentation of their ideas for a future resubmission.